# Novel and Potential Small Molecule Scaffolds as DYRK1A Inhibitors by Integrated Molecular Docking-Based Virtual Screening and Dynamics Simulation Study

**DOI:** 10.3390/molecules27041159

**Published:** 2022-02-09

**Authors:** Mir Mohammad Shahroz, Hemant Kumar Sharma, Abdulmalik S. A. Altamimi, Mubarak A. Alamri, Abuzer Ali, Amena Ali, Safar Alqahtani, Ali Altharawi, Alhumaidi B. Alabbas, Manal A. Alossaimi, Yassine Riadi, Ahmad Firoz, Obaid Afzal

**Affiliations:** 1Department of Pharmaceutical Chemistry, College of Pharmacy, Sri Satya Sai University of Technology and Medical Sciences, Sehore 466001, Madhya Pradesh, India; mirshahroz@gmail.com; 2Department of Pharmaceutical Chemistry, College of Pharmacy, Prince Sattam Bin Abdulaziz University, P.O. Box 173, Al Kharj 11942, Saudi Arabia; as.altamimi@psau.edu.sa (A.S.A.A.); m.alamri@psau.edu.sa (M.A.A.); safar.alqahtani@psau.edu.sa (S.A.); altharawi@gmail.com (A.A.); ab.alabbas@psau.edu.sa (A.B.A.); m.alossaimi@psau.edu.sa (M.A.A.); y.riadi@psau.edu.sa (Y.R.); 3Department of Pharmacognosy, College of Pharmacy, Taif University, P.O. Box 11099, Taif 21944, Saudi Arabia; abuali@tu.edu.sa; 4Department of Pharmaceutical Chemistry, College of Pharmacy, Taif University, P.O. Box 11099, Taif 21944, Saudi Arabia; amrathore@tu.edu.sa; 5Department of Biological Sciences, Faculty of Science, King Abdulaziz University, P.O. Box 80200, Jeddah 21589, Saudi Arabia; ahmadfirozbin@gmail.com

**Keywords:** alzheimer’s disease (AD), DYRK1A, kinases, molecular docking, molecular dynamics (MD) simulation, ProTox-II, virtual screening

## Abstract

The dual-specificity tyrosine phosphorylation-regulated kinase 1A (DYRK1A) is a novel, promising and emerging biological target for therapeutic intervention in neurodegenerative diseases, especially in Alzheimer’s disease (AD). The mol*Mall* database, comprising rare, diverse and unique compounds, was explored for molecular docking-based virtual screening against the DYRK1A protein, in order to find out potential inhibitors. Ligands exhibiting hydrogen bond interactions with key amino acid residues such as Ile165, Lys188 (catalytic), Glu239 (gk+1), Leu241 (gk+3), Ser242, Asn244, and Asp307, of the target protein, were considered potential ligands. Hydrogen bond interactions with Leu241 (gk+3) were considered key determinants for the selection. High scoring structures were also docked by Glide XP docking in the active sites of twelve DYRK1A related protein kinases, viz. DYRK1B, DYRK2, CDK5/p25, CK1, CLK1, CLK3, GSK3β, MAPK2, MAPK10, PIM1, PKA, and PKCα, in order to find selective DYRK1A inhibitors. MM/GBSA binding free energies of selected ligand–protein complexes were also calculated in order to remove false positive hits. Physicochemical and pharmacokinetic properties of the selected six hit ligands were also computed and related with the proposed limits for orally active CNS drugs. The computational toxicity webserver ProTox-II was used to predict the toxicity profile of selected six hits (mol*mall* IDs 9539, 11352, 15938, 19037, 21830 and 21878). The selected six docked ligand–protein systems were exposed to 100 ns molecular dynamics (MD) simulations to validate their mechanism of interactions and stability in the ATP pocket of human DYRK1A kinase. All six ligands were found to be stable in the ATP binding pocket of DYRK1A kinase.

## 1. Introduction

Alzheimer’s disease (AD) is distinguished by irremediable neurodegeneration as well as diminishing cognitive functions. About 60–70% cases of dementia is attributed to AD [1]. There are approximately 50 million patients of dementia worldwide, and this number is added by nearly 10 million each year [2]. The currently available symptomatic treatments for AD comprise mainly acetylcholinesterase inhibitors especially donepezil and rivastigmine [3]. These drugs are incapable to stop the progression of this disease. As on 7 June 2021, US FDA (Food and Drug Administration) approved first-of-its-kind treatment for AD, a drug Aduhelm (Aducanumab, a monoclonal antibody), claimed to reduce β-amyloid (Aβ) plaque in the brain. It is developed jointly by Biogen Inc. (Cambridge, MA, USA) and Eisai Co. Ltd., (Tokyo, Japan), with $56,000 annual cost burden on patients [4]. The Aβ plaque hypothesis was suggested as a predominant exposition for the adverse neurological cascades in AD [5]. The existence of insoluble extracellular Aβ plaques, and insoluble intracellular neurofibrillary tangles (NFTs) in the brain are the characteristic features of AD [6]. Decades of research on AD have recognized novel drug targets for the therapeutic intervention, the most promising among them is DYRK1A (dual-specificity tyrosine phosphorylation-regulated kinase 1A) [7,8,9,10].

DYRK1A is referred to as dual specificity kinase, because of its ability of self-activation by autophosphorylation of Tyr321 in its activation loop, as well as phosphorylation of a wide variety of exogenous protein substrates [11]. It is one of the 5 subtypes (1A, 1B, 2, 3, and 4), DYRK1B and DYRK2 being the closely related homologues [12]. Although found ubiquitously in the brain, DYRK1A is shown to be overexpressed in regions such as hippocampus, olfactory bulb, and cerebellum, during the early embryonic developmental stages [13]. The gene responsible for DYRK1A is found in the Down Syndrome Critical Region (DSCR) of human chromosome 21. Its overexpression and, hence, its amplified activity, was reported in patients suffering from neurodegenerative diseases. In the frontal cortex, DYRK1A positive nuclei were found to be approximately 20 times higher in AD as compared to normal brains [14,15]. DYRK1A directly and indirectly promotes pathogenesis of AD by ultimately supporting the formation of NFTs [16,17,18,19,20] and neurotoxic Aβ plaques [21,22,23]. Inhibition of DYRK1A function also alleviates several other pathways responsible for the development of neurodegeneration in AD [14,15]. 

A substantial proportion of works attempting to identify potent and selective DYRK1A inhibitors has been reported over the past few decades (Figure 1) [24,25,26,27,28,29,30]. Additionally, multiple DYRK1A-inhibitor co-crystallized structures were reported, providing vital information about the ATP binding environment, essential for the development of selective inhibitors. Epigallocatechin gallate (EGCg), a polyphenol and catechin of green tea, was discovered as a nonselective but somewhat potent DYRK1A inhibitor (IC_50_ 330 nM) [31]. Harmine (a β-carboline alkaloid) was reported as a potent and moderately selective inhibitor of DYRK1A (IC_50_ 33 nM) [32]. Harmine (denoted as HRM in PDB ID: 3ANR) was shown to bind in the ATP binding space, in which the pyridine ring is involved in hydrophobic pi-pi interaction with Phe238, pyridine nitrogen formed hydrogen bond with Lys188, and the –OCH_3_ group formed another hydrogen bond with Leu241 [33]. Harmine inhibits tau phosphorylation by DYRK1A at Ser396 (IC_50_ 700 nM), while analogues such as harmol and 9-ethylharmine showed better IC_50_ values of 90 and 400 nM, correspondingly [18]. However, *β*-carboline analogues cannot be explored therapeutically due to their significant drawbacks [34]. INDY (a benzothiazole derivative), a DYRK1A/CLK dual inhibitor (DYRK1A, IC_50_ 0.24 μM, [ATP] 10 μM), was obtained by the structural modification of TG003 (DYRK1A IC_50_ 0.93 μM; CLK1 IC_50_ 119 nM; CLK4 IC_50_ 30 nM) [33]. INDY (denoted by EHB in PDB ID: 3ANQ) was shown to bind in parallel fashion such as harmine, where the O of the –OH group forms a H-bond with Leu241, and the O of the carbonyl function forms another H-bond with Lys188 [33]. A quinazoline amine derivative was reported as a strong inhibitor of DYRK1A (IC_50_ 14 nM) [35]. A benzocoumarin (dNBC) derivative was described as a good inhibitor of DYRK1A (IC_50_ 0.60 μM, [ATP] 20 μM) [36]. A lamellarin D analogue (chromeno [3,4-*b*]indole derivative), exhibited potent DYRK1A inhibition with an IC_50_ 67 nM [37]. Leucettine L41 in complex with human DYRK1A (denoted by 3RA in PDB ID: 4AZE, Kd 7.8 nM), reported as an ATP competitive inhibitor, formed two direct polar interactions, similar to harmine and INDY, with Leu241 and Lys188 [38]. The co-crystallized structure of an indazole derivative (denoted as D15 in PDB ID: 2WO6) indicated that the indazole moiety interacts via two H-bonds with Leu241 and Glu239, along with a salt bridge that forms between the primary amine and the carboxylate side chain of Asp307 [39]. An indolo[3,2-*c*]quinoline-6-carboxylic acid analogue (4E2 in PDB ID: 4YLK) revealed the interactions in the ATP binding space of DYRK1A (IC_50_ 6 nM), via (i) a salt bridge amid the carboxylate group of 4E2 and the amine of Lys188, (ii) water interceded hydrogen bonds with Ser242 & Asp307, and (iii) flipped P-loop residue Phe170 forming pi-pi hydrophobic interactions with the inhibitor. In addition, the iodine at position 10 was found located nearby gatekeeper (gk) Leu241 at a distance of 3.3 Å [40]. A hydroxyl acetamido benzothiazole (denoted as QIV in PDB ID: 5A3X) revealed active site binding (DYRK1A IC_50_ 400 nM) with the acetamide group forming hydrogen bond to the catalytic Lys188, and the 5-OH group forming hydrogen bond with the gk Leu241 [41]. Czarna et al., screened their in-house library of 1000 compounds, and reported 9 potential DYRK1A inhibitors (Ki values in the range of 104 to 1680 nM) with different core structures [42]. Weber et al., recently reported several 2-methyl-3*H*-imidazo[4,5-*b*]pyridin-5-yl]pyridine-2,6-diamine derivatives, among them the most promising and selective DYRK1A inhibitor has an IC_50_ value of 5 nM (DYRK2 IC_50_ = 195 nM) [43].

The gene for DYRK1A is found to be located within DSCR on human chromosome 21, and studies revealed that upregulation of DYRK1A is the key factor responsible for cognitive decline in patients of AD and DS [14,15]. Since current symptomatic treatments for cognitive deficiencies are limited and inefficient, inhibition of DYRK1A function in the brain by small molecules offers a good opportunity for pharmaceutical interference for neurodegeneration associated with AD and DS. In this report, we identified some novel and selective potential DYRK1A inhibitors with different structural scaffolds through a comprehensive molecular docking-based virtual screening approach, validated by molecular dynamics simulations. The identified inhibitors were found to have a good in silico physicochemical, pharmacokinetic and toxicity profile. The workflow adopted to identify novel, potent and selective DYRK1A inhibitors is depicted in Figure 2.

## 2. Results

### 2.1. Docking Library

The downloaded mol*Mall* database containing 15,381 chemically diverse structures was checked for any duplicates by using OpenBabel (ver. 3.0.0) and found to be unique without any duplicates. It was prepared for docking-based virtual screening by the application LigPrep (ver. 3.4). The resulting docking library consisting of 54,594 structures was utilized for comprehensive molecular docking-based virtual screening against a set of DYRK1A protein kinases.

### 2.2. Human DYRK1A and Related Protein Kinases

In order to increase the accuracy of the results, five DRYK1A protein crystal structures (PDB IDs: 3ANQ, 4AZE, 2WO6, 4YLK and 5A3X), available in the protein data bank (PDB) from diverse reputed laboratories and reported in different timelines as having different resolutions, were used for docking-based virtual screening. The difference in the crystal structure of these five DYRK1A kinases were visualized by superimposing with the help of the PyMOL Molecular Graphics System (v1.8.4.0). A slight variation near αC helix and β1, β2 and β3 sheets were observed among the crystal structures and is depicted in Figure 3. The ATP binding space in DYRK1A kinase protein, essential for the development of selective inhibitors, is a conserved region and consists of Phe238 (gatekeeper; gk), Glu239 (gk+1), Leu241 (gk+3), Lys188 (catalytic), Phe170, Ser242, Asn292, and Asp307 amino acid residues [33,38,39,40,41]. In order to find out selective DYRK1A inhibitors, the potent and common ligands obtained after docking-based virtual screening were also screened for their binding affinity with the other twelve related protein kinases [PDB IDs: 4D2S (DYRK1B), 3KVW (DYRK2), 3O0G (CDK5/p25), 5W4W (CK1), 1Z57 (CLK1), 3RAW (CLK3), 5OY4 (GSK3β), 2PZY (MAPK2), 3RTP (MAPK10), 5VUA (PIM1), 5LCP (PKA), and 1DSY (PKCα)].

### 2.3. Validation of the Docking Protocol

Each of the co-crystallized ligands, viz. EHB (of 3ANQ), 3RA (of 4AZE), D15 (of 2WO6), 4E2 (of 4YLK), and QIV (of 5A3X) of the DYRK1A protein kinase, were docked into the ATP pocket of their corresponding pdb structures by Glide extra precision (XP) docking. The docked ligand’s top poses based on their scores and interactions with the active site residues were aligned with the co-crystallized poses in order to validate the docking protocol. The heavy atom RMSD for EHB, 3RA, D15, 4E2, and QIV were found to be 0.742 Å, 0.932 Å, 0.978 Å, 0.573 Å and 0.466 Å, respectively (Figure 4). The docking protocols that give conformations below a preselected value from the known poses (usually 1.5 or 2 Å depending on ligand size) are generally considered to have been implemented effectively [44].

The docking protocol was further validated by area under receiver operating characteristic (ROC) curves. Five active DYRK1A inhibitors (INDY, a lamellarin D analogue, a quinazoline amine derivative, Leucettine L41, and harmine) were used to generate 213 decoys (inactive inhibitors) by using DUDE webserver (A Database of Useful Decoys: Enhanced) [45]. All the 218 ligands were docked into the ATP binding pocket of five DYRK1A structures having PDB IDs viz. 2WO6, 3ANQ, 4AZE, 4YLK and 5A3X. The obtained results with docking scores of 218 ligands for each target structures were provided as an input file for the generation of ROC curves by the screening explorer webserver [46]. The ROC-AUC were found to be 0.948, 0.960, 0.975, 0.917 and 0.957, respectively for the five DYRK1A structures having PDB IDs 2WO6, 3ANQ, 4AZE, 4YLK and 5A3X. The ROC curve is depicted in Figure 5.

### 2.4. Docking-Based Virtual Screening

The library of compounds comprising 54,594 structures, prepared and obtained from LigPrep application, were exposed to docking-based virtual screening using the generated grid of five DYRK1A receptors, having PDB IDs viz. 3ANQ, 4AZE, 2WO6, 4YLK and 5A3X. HTVS with 1% output, SP docking with 10% output, and XP docking with 100% output options were selected. As a result, a total of 54 molecules were obtained from each subjected protein structure after docking-based virtual screening. A total of 41 ligand structures were found to be common in all output results. Out of these 41 molecules, a bis-indole indigoid scaffold (mol*Mall* ID: 19394) was already reported previously as a DYRK1A inhibitor, so this molecule was not included in further study [47]. A total of 40 structures were retained for further binding pattern analysis. Glide XP docking scores of the six final identified hit molecules and three known DYRK1A inhibitors is provided in Table 1. Molecular contacts profiling for the identified six hit molecules and three known inhibitors in the selective ATP pocket of human DYRK1A (PDB ID: 3ANQ) is provided in Table 2.

### 2.5. Analysis of the Binding Pattern

The selected 40 ligand structures were selected for their binding pose investigation inside the ATP binding pocket of DYRK1A structures. Ligands having hydrogen bond interactions with key amino acid residues such as Ile165, Lys188 (catalytic), Glu239 (gk+1), Leu241 (gk+3), Ser242, Asn244, and Asp307 of the target protein were retained. Hydrogen bond interactions with Leu241 (gk+3) were considered key determinants for the selection and those having no interaction with the same were rejected [41]. Three-dimensional binding interactions (surface and cartoon view) and 2D binding pattern diagrams of the six final selected structures, along with reported known inhibitors (EHB, Harmine and EGCG), are depicted in Figure 6, Figure 7 and Figure 8, respectively.

### 2.6. MM/GBSA Binding Free Energy Calculations

Binding free energies (ΔG_bind_) of selected ligand–protein complexes were calculated in order to remove false positive hits. A total of 32 ligands having ΔG_bind_ less than −40 kcal/mol, and high Glide XP score less than −7.0, were retained for further docking with related DYRK1A related protein kinases in order to obtain selective inhibitors. The selection criteria were set keeping in mind the docking scores and ΔG_bind_ values of two reported DYRK1A inhibitors, EHB and Harmine. Prime MM/GBSA binding free energy (kcal/mol) of the six final identified hit molecules and three known DYRK1A inhibitors is provided in Table 1.

### 2.7. Glide XP Docking with Related Protein Kinases

The selected 32 hits were docked by Glide XP docking in the ATP pocket of 12 DYRK1A-related protein kinases viz. DYRK1B, DYRK2, CDK5/p25, CK1, CLK1, CLK3, GSK3β, MAPK2, MAPK10, PIM1, PKA, and PKCα in order to find the selectivity of the hit ligands [48,49,50,51,52,53,54,55,56]. Ligands with a Glide XP score less than −5.0 and Prime MM/GBSA ΔG_bind_ below −40 kcal/mol were rejected and considered to be non-selective. Among them, the compound Quercetin (mol*Mall* IDs: 22067) was found to have a high Glide XP score and binding free energy, with 10 out of 12 kinases. A total of 15 structures were retained for further analysis of their physicochemical and pharmacokinetic properties. Glide XP scores and Prime MM/GBSA binding free energy (kcal/mol) of the six final identified hit molecules obtained after docking with related 12 protein kinases is provided in Table 3. The heatmap analysis of the binding energy of six identified hits with all the studied kinases (DYRK1A and related kinases) are depicted in Figure 9. Among the identified hits, molecules with mol*mall* ID 15938 were found to have inhibitory potential towards DYRK2, GSK3β and MAPK10 kinases, while 11352 showed some affinity towards DYRK2 kinase.

### 2.8. Physicochemical and Pharmacokinetic Properties

Ligands showing great variations from the qualifying limits for orally active CNS drugs were discarded [57]. Among them, the noticeable compounds with mol*Mall* IDs: 15677 (6,6′-dimethoxy-3,4′-biisoquinoline-7,7′-diol), 10992 ((1Z)-1*H*-naphtho[2,3-*e*]indole-1,2(3*H*)-dione-1-(phenylhydrazone)), 20271 (4-phenyl-1-(3a,4,7,7a-tetrahydro-1*H*-4,7-methanoinden-1-yl)-1,2,4-triazolidine-3,5-dione), and 18976 (2-hydroxy-4-(4-hydroxyphenyl)-1*H*-phenalen-1-one). Structures having mol*Mall* IDs: 15938 and 19037, were found to have slight variation from the qualifying limits in terms of FOSA and PISA only, and thus were retained. A total of six structures were retained for further studies. The QP properties of the final six selected identified hit molecules are presented in Appendix A.

### 2.9. Toxicity Studies

A preliminary evaluation of the toxicity viz. acute oral toxicity (LD_50_), hepatotoxicity, mutagenicity, carcinogenicity, cellular toxicity, immunotoxicity, and toxicological pathways (nuclear receptor signalling and stress response pathways) of a chemical compound is of great importance in the discipline of drug discovery. In silico toxicity models can accurately predict the toxicity effects of chemicals and thus minimize time, costs and the need for animal testing. The computational toxicity webserver ProTox-II [58] was utilized to predict the toxicity profile of the six selected hits, along with the prediction accuracy, and are provided in Appendix A. Compounds having mol*Mall* IDs: 21830, 11352 and 21878 were found to be inactive and safe in terms of all the toxicological parameters. Compounds with mol*Mall* ID: 15938 were predicted to be carcinogenic (64% probability) and mutagenic (54% probability). Similarly, compounds with mol*Mall* ID: 19037 were predicted to be hepatotoxic (59% probability) and carcinogenic (72% probability). Compounds with mol*Mall* ID: 9539 were found to be the most toxic, with the prediction accuracy of 54.26% in terms of acute oral toxicity (LD_50_ 1 mg/kg), hepatotoxicity (56% probability) and carcinogenicity (55% probability). Despite the toxic probability of the three hits, all six of the final hits were included for further MD simulation study in order to confirm their stability in the ATP pocket of DYRK1A kinase, and indicate the probability of structural alterations in the scaffolds to find safer leads.

### 2.10. Molecular Dynamics (MD) Simulation

MD simulations were used to assess the interaction mechanism and stability of putative ligands in the ATP binding pocket of human DYRK1A kinase, using six docked ligand–protein complexes, with reference to the co-crystallized structure (EHB). The RMSD, Rg, RMSF, SASA, potential binding energy, hydrogen bond analysis, and PCA of the docked complexes were evaluated.

The simulated system’s time to achieve structural equilibrium was calculated using RMSD. This is a crucial calculation that determines how a protein’s molecular structure varies or changes. For the length of a 100 ns MD simulation, the equilibrium time for simulated protein–ligand complexes were determined using the backbone RMSD. Figure 10A shows the RMSD values of protein backbones that were computed. After 7.5 ns, all systems were equilibrated, and a substantial deviation in the protein backbone was seen during the first 10 ns simulation, which was predicted, given the abrupt shift in the protein environment. Complexes were then stabilized and demonstrated a constant state of dynamic behaviour for the period of the 100 ns simulation duration, with a tiny overall fluctuation range of RMSD 6.8 to 7.8 nm. With regard to the reference complex (EHB), all the complexes displayed a nearly identical RMSD fluctuation pattern. The RMSF parameter determines how much a residue contributes to a complicated structure’s stability. The RMSF measurements revealed minor variations in the protein’s backbone, with an average value of less than 0.5 nm (Figure 10B). Most of the disrupted residues in the reference complex are in the loop regions, away from the ligand-binding pocket location.

Rg was calculated by measuring the molecular volume and density of protein. The root mean square distance between each atom in the system and its centre of mass is the radius of gyration. All of the protein–ligand complexes (EHB, 9539, 11352, 15938, 19037, 21830, and 21878) had steady Rg values between 2.21 and 2.31 nm, indicating protein stability, as the low values obtained over the 100 ns simulation duration suggested protein compactness and stable inhibitor binding (Figure 11A). The retention of stable conformation of smaller molecules inside the active pocket of the protein is dependent on hydrogen bonding [59]. Figure 11B shows H-bonds for all systems during the course of 100 ns of simulation time. The ligands 11352, 15938, 21830, and 21878 are the most promising, as shown in the figure, since they interacted with a greater average number of H-bonds (up to 4) than the reference, EHB.

The SASA was also calculated to evaluate the constancy of simulated systems, and to determine how much of the receptor area was exposed to the solvents. A larger SASA value represents the growth of protein volume during MD simulation. The interaction of the ligands in the active site can change SASA and protein folding [60]. The calculated SASA values observed for the ligands were between 188 and 196 nm^2^, reflecting that the binding does not affect the protein structure (Figure 12A), and also suggesting that complexes were stable after the binding of ligands inside the active site. The potential binding energy significantly contributes to the molecular interaction between the ligands and protein. The higher negative values of the binding energy reflected that the targeted compound favourably interacted within the active site of the receptor. Ligands 21830 and 21878 are the most promising ones, since they showed higher negative binding energy (Figure 12B).

Essential dynamics (ED) or principal component analysis (PCA) is a reliable approach for classifying protein conformations and identifying massive, coordinated patterns of fluctuations from MD simulation trajectories. The PCA score plot (Figure 13) revealed different clusters formations. Among them, 21830 (green) and 11352 (golden) are overlapped. The other complex exhibited significant differences by forming a distinct cluster.

### 2.11. In Silico Bioactivity Prediction

To predict the bioactivity, the structure of the identified six hits and reported three inhibitors were submitted to the Swiss Target Prediction webserver (http://www.swisstargetprediction.ch/; accessed on 23 January 2022) to determine their kinase inhibitory potential, provided in Appendix A. The co-crystallized ligand EHB showed the highest % kinase inhibition (86.7%) and the webserver accurately predicted the target as DYRK1A. This accurate prediction can be attributed to the presence of co-crystallized DYRK1A kinase structure (PDB ID: 3ANQ) in their database. However, for the rest two reported inhibitors of DYRK1A, harmine and EGCG, the % kinase inhibition was found to be 26.7% and 13.3%, respectively. This can be attributed to their non-selective nature. Among the identified hits, ligands having mol*mall* ID 11352 and 9539 predicted to have the highest % kinase inhibition, 66.7% and 40%, respectively. The molinspiration webserver (https://www.molinspiration.com/cgi-bin/properties; accessed on 23 January 2022) was also used to predict the kinase inhibition potential of the molecules. A larger value of bioactivity score refers to the higher probability of the molecule being active. Ligands with mol*mall* ID 11352 were predicted to have the highest kinase bioactivity score (+0.50), followed by harmine (+0.31). However, the bioactivity score for the co-crystallized ligand EHB was found to be −0.47 (Appendix A).

### 2.12. Structure Similarity Comparison

The structure similarity of the six identified hits were compared with the co-crystallized ligand structures of DYRK1A and related kinases by ChemMine Web Tools (https://chemminetools.ucr.edu/; accessed on 23 January 2022) to study the structure activity relationship, since similar structures have similar biological functions. The distance matrix (Z-scores) was computed and provided in Appendix A. The heatmap visualization of the distance matrix (Z-scores) was prepared by using the GraphPad Prism (version 9.1.0), depicted in Figure 14. The same Z-scores were obtained for kinase, DYRK1A (PDB ID: 4AZE) and CLK3 (PDB ID: 3RAW), due to the presence of the same co-crystallized ligand, 3RA, in their ATP binding pocket. Molecules having mol*mall* ID 21830 and 19037 were found to be more structurally similar to the co-crystallized ligand 3RA of DYRK1A (4AZE), while 11352 and 9539 showed maximum structure similarity with 4E2 of DYRK1A (4YLK). Molecule 15938 showed the highest structure similarity with ligands 3RA and D15 of DYRK1A among all the co-crystallized ligands studied. Molecule 21878 showed greater similarity with the ligand DBQ of CLK1 (1Z57), followed by 3RA of DYRK1A. Overall, from the results, it was concluded that the identified hits have the potential to inhibit other kinases such as DYRK2, CLK1 and CLK3, besides DYRK1A.

### 2.13. Synthetic Accessibility (SA) Prediction

The SyntheticAccessibiliyCli (Ambit-SA) software tool is a java program that computes synthetic accessibility (SA) scores for the given molecules ranging from 0 to 100. The value 100 denotes maximal synthetic accessibility and, hence, the molecule is easily synthesizable. This program includes molecular, stereo-chemical, and fused and bridged system complexities of the molecule and calculates SA scores on the basis of four molecular descriptors, representing different structural and topological features. Five out of six identified molecules were found to have satisfactory SA scores and are predicted to be easily synthesizable. The least SA score was found for the molecule having mol*mall* ID 15938 (Table 4).

## 3. Discussion

As most of the previously discovered small molecules targeting β-secretase and γ-secretase, have been failed in clinical trials, discovery of selective and potent DYRK1A inhibitors is an emerging research area for AD treatment [61,62,63,64]. In this study, six novel and selective potential DYRK1A inhibitors with diverse structural scaffolds were identified by screening mol*Mall* database, consisting of rare, diverse and unique compounds, available commercially. The prepared mol*Mall* docking library, consisting of 54,594 structures, was utilized for molecular docking-based virtual screening against a set of five DYRK1A protein kinase crystal structures (PDB IDs: 3ANQ, 4AZE, 2WO6, 4YLK and 5A3X) in order to increase the accuracy of the results. As a result, a total of 54 small molecules were obtained from each subjected protein structure. A total of 41 ligand structures were found to be common in output results. Out of these 41 molecules, a bis-indole indigoid scaffold (mol*Mall* ID: 19394) was already reported previously, as a DYRK1A inhibitor [47]. A total of 40 structures were retained for further binding pattern analysis. The ATP binding pocket in DYRK1A kinase, essential for the binding of inhibitors, is a conserved region consisting of Phe238 (gatekeeper; gk), Glu239 (gk+1), Leu241 (gk+3), Lys188 (catalytic), Phe170, Ser242, Asn292, and Asp307 amino acid residues [33,38,39,40,41]. After MM/GBSA binding energy calculations, l of 32 ligands having ΔG_bind_ less than −40 kcal/mol were retained and selected for further study. Identification of a potent and selective kinase inhibitor is a hectic and exciting task, since a large number of protein kinases share the same ATP binding site environment [65]. To find out a selective DYRK1A inhibitors, potent and common 32 ligands obtained after docking-based virtual screening were screened for their binding affinity with the other twelve related protein kinases [PDB IDs: 4D2S (DYRK1B), 3KVW (DYRK2), 3O0G (CDK5/p25), 5W4W (CK1), 1Z57 (CLK1), 3RAW (CLK3), 5OY4 (GSK3β), 2PZY (MAPK2), 3RTP (MAPK10), 5VUA (PIM1), 5LCP (PKA), and 1DSY (PKCα)]. Ligands with a Glide XP score less than −5 and Prime MM/GBSA binding free energy below −40 kcal/mol were rejected and considered to be non-selective. Among them, the compound Quercetin (mol*Mall* IDs: 22067) was found to have a high Glide XP score and binding free energy, with 10 out of 12 kinases. Various physicochemical and pharmacokinetic attributions (QP properties) of the selected 15 hit ligands were computed and equated with the eligibility limits as recommended for orally active CNS drugs [57]. The computational toxicity webserver ProTox-II [58] was utilized to predict the toxicity profile of the selected potential six hits, having mol*Mall* IDs: 9539, 11352, 15938, 19037, 21830, and 21878. Despite the slight toxic probability of the three hits, all the six final hits were included for further MD simulation study, which confirmed their stability in the ATP pocket of DYRK1A kinase. Among the selected six hits, molecules having mol*mall* ID 15938, 11352 and 21878 were also found to have affinity towards DYRK2, GSK3β, CLK1, CLK3 and MAPK10 protein kinases, in addition to DYRK1A. The molecules were predicted to be easily synthesizable. Therefore, their synthesis and kinase profiling are further warranted to establish them as selective DYRK1A inhibitors.

## 4. Materials and Methods

### 4.1. Preparation of the Docking Library

The mol*Mall* database, (MDPI database; URL: http://www.molmall.net/download.html; accessed on 16 September 2017), with 15,381 diverse chemical structures, was downloaded in SDF format and imported into Maestro (version 10.2), the graphical user interface (GUI) of Schrödinger computational software [66]. The application LigPrep (ver. 3.4) was used for the preparation of the docking library in maestro format, which includes the generation of a three-dimensional (3D) structure, energy minimization using OPLS 2005 force field, ionization at pH 7.0 ± 2.0 using ionizer, and the generation of the tautomers and stereoisomers.

### 4.2. DYRK1A and Related Protein Kinases: Selection, Preparation and Grid Generation

Five DYRK1A (PDB IDs: 3ANQ, 4AZE, 2WO6, 4YLK and 5A3X) and twelve related protein kinases [PDB IDs: 4D2S (DYRK1B), 3KVW (DYRK2), 3O0G (CDK5/p25), 5W4W (CK1), 1Z57 (CLK1), 3RAW (CLK3), 5OY4 (GSK3β), 2PZY (MAPK2), 3RTP (MAPK10), 5VUA (PIM1), 5LCP (PKA), and 1DSY (PKCα)] X-ray co-crystallized structures were downloaded from the RCSB Protein Data Bank (PDB) website (URL: http://www.rcsb.org/; accessed on 25 October 2017) in PDB format [33,38,39,40,41,48,49,50,51,52,53,54,55,56]. Protein structures in PDB format were imported on Maestro workspace and prepared for docking by protein refinement tools available in workflows option. Protein structures were pre-processed and analysed. Bond orders were corrected, H atoms were appended, and H_2_O molecules, including any heteroatoms other than co-crystallized ligands, were excised. The H-bonding network in the protein structures were optimized by exhaustive sampling option and then, finally, the energy of the optimized structures was minimized to RMSD at 0.3 Å by Impref minimization option by using force field OPLS 2005. The refined and energy minimized protein structures in maestro format were used for the receptor grid (20 × 20 × 20 Å) generation to define the docking site by identifying the co-crystallized ligands, through Receptor Grid Generation of Glide available on applications panel. The grid file for each protein structure was subsequently exposed for docking.

### 4.3. Validation of the Docking Protocol

The 2D structures of each co-crystallized ligand, viz. EHB (3ANQ), 3RA (4AZE), D15 (2WO6), 4E2 (4YLK), QIV (5A3X), DYK (4D2S), IRB (3KVW), 3O0 (3O0G), 9WG (5W4W), DBQ (1Z57), 3RA (3RAW), B4K (5OY4), B18 (2PZY), 34I (3RTP), 8GX (5VUA), M77 (5LCP), and PSF (1DSY), were drawn on ChemDraw Ultra (ver. 12.0) and saved in SDF format. These 2D structures in SDF format were imported into Maestro workspace and processed by the application LigPrep by the method discussed above. The prepared structures were docked into the ATP pocket of their corresponding receptor by Glide extra precision (XP) docking [67,68]. The docked ligand exhibiting orientation similar to that of the co-crystallized ligands were saved and superimposed with the co-crystallized poses, in order to check the accuracy of the docking protocol.

Furthermore, the docking protocol was also validated by area under receiver operating characteristic (ROC) curves. Five known and potent DYRK1A inhibitors, namely INDY, a lamellarin D analogue, a quinazoline amine derivative, Leucettine L41, and harmine (Figure 1), were selective as a set of active inhibitors. These active inhibitors were used to generate 213 decoys (inactive inhibitors) online by using DUDE webserver (A Database of Useful Decoys: Enhanced) [45]. The decoys were enriched with the actives and all the 218 ligands were prepared and docked into the ATP binding pocket of five DYRK1A structures having PDB IDs viz. 2WO6, 3ANQ, 4AZE, 4YLK and 5A3X. The obtained CSV result files for each DYRK1A crystal structures having the docking scores were provided as an input file for the generation of ROC curves online by the webserver, screening explorer [46].

### 4.4. Docking-Based Virtual Screening

Virtual screening workflow available in Schrödinger computational software [66] was used for docking-based virtual screening of the library comprising 54,594 structures, using the generated grid of five DYRK1A receptors, having PDB IDs viz. 3ANQ, 4AZE, 2WO6, 4YLK and 5A3X. For the virtual screening, HTVS (high-throughput virtual screening) with 1% output, SP (standard precision) with 10% output, and XP (extra precision) with 100% output docking mode options were selected. The common top scoring ligands obtained after virtual screening were saved in pdb format, along with their corresponding proteins for their binding pattern analysis.

### 4.5. Analysis of the Binding Pattern

The ligand–protein complexes of top scoring ligands were analysed by Biovia Discovery Studio Visualizer (v20.1.0.19295) [69]. Molecules having hydrogen bond interactions with key amino acid residues, such as Ile165, Lys188, Glu239, Leu241, Ser242, Asn244, and Asp307 of the target protein, were retained for binding free energy calculations. Two-dimensional binding pattern diagrams were obtained and saved for the selected compounds by Discovery Studio Visualizer. Three-dimensional binding interaction pictures (surface and cartoon view) for the docked ligands were obtained from PyMOL Molecular Graphics System (v1.8.4.0, Schrödinger Inc., New York, NY, USA) [70].

### 4.6. MM/GBSA Binding Free Energy Calculations

Prime (v4.0) MM-GBSA modules in Maestro–Schrödinger suite 2015.2 was utilized for the estimations of binding free energies (ΔG_bind_) of selected ligand–protein complexes, using default parameters (OPLS_2005 force field and VSGB solvent model), to remove false positive hits [71]. A more negative ΔG_bind_ value designates durable interaction. The heatmap visualization of the binding energy of the six identified hits with all the studied kinases (DYRK1A and related kinases) was obtained by using GraphPad Prism (version 9.1.0).

### 4.7. Glide XP Docking with Related Protein Kinases

Selected structures were further docked by Glide XP docking by the method reported above in the active sites of twelve DYRK1A related protein kinases [PDB IDs: 4D2S (DYRK1B), 3KVW (DYRK2), 3O0G (CDK5/p25), 5W4W (CK1), 1Z57 (CLK1), 3RAW (CLK3), 5OY4 (GSK3β), 2PZY (MAPK2), 3RTP (MAPK10), 5VUA (PIM1), 5LCP (PKA), and 1DSY (PKCα)] in order to find the selectivity of the hit ligands [48,49,50,51,52,53,54,55,56].

### 4.8. Physicochemical and Pharmacokinetic Properties

The QikProp 4.4 module of Maestro–Schrödinger suite 2015.2 was utilized for the estimation of thirty-five QikProp (QP) attributions of the selected ligands [72]. Chemical structures of the ligands were neutralized preceding to the estimation of QP attributions. The attributions were predicted and compared with the qualifying limits as proposed for orally active CNS drug discovery [57].

### 4.9. Toxicity Studies

The computational toxicity webserver ProTox-II was used to predict and compute the toxicity profile of selected hits [58]. Acute toxicity (oral, LD_50_), hepatotoxicity, mutagenicity, carcinogenicity, cellular toxicity, immunotoxicity, and toxicological pathways (nuclear receptor signalling and stress response pathways) were computed, along with the accuracy of the prediction.

### 4.10. Molecular Dynamics (MD) Simulation

MD simulations were used to confirm the interactions, constancy, and activity of recognised ligand structures in the ATP pocket of human DYRK1A protein (PDB ID: 3ANQ), with reference to DYRK1A structure, co-crystallized with a benzothiazole derivative, EHB [33]. GROMACS 2018.1 package [73,74] and the CHARMM36 all-atom force field were utilized for 100 ns MD simulations of chosen docked complexes [75]. Using the CHARMM General Force Field (CGenFF) tool, the topology and parameter files for the ligands were created [76]. After that, each docked complexes were solvated in an orthorhombic box using the four-point TIP4P water model [77] at a distance of at least 1 nm from the docked ligand and protein system. The system was neutralized and counterbalanced by adding nine chloride (Cl^−^) ions. Furthermore, the ionic strength of the solution of the system was maintained by adding 0.15 M NaCl to realistically mimic the physiological conditions. During the MD simulation, periodic boundary conditions (pbc) were used. For energy minimization, a steepest descent method with a maximum step size of 0.01 nm and a tolerance of 1000 kJ/mol/nm was utilized. The Linear Constraint Solver (LINCS) method was utilized to limit bond lengths. The particle mesh Ewald (PME) technique was used to conduct electrostatic calculations. The system was equilibrated utilizing canonical ensembles NVT, followed by isothermal-isobaric ensemble (NPT) for 100 ps after energy reduction. All simulations were run at the same temperature and pressure (300 K, 1 atm). Trajectories were created every 2 fs and stored every 2 ps during the manufacturing run, which lasted 100 ns. GROMACS analysis programs were used to perform all preliminary analyses, comprising root mean square deviation (RMSD), radius of gyration (Rg), root mean square fluctuations (RMSF), potential binding energy, solvent-accessible surface area (SASA), hydrogen bond (H-bond), and principal component analysis (PCA).

### 4.11. In Silico Bioactivity Prediction

The identified six hits and reported three inhibitors were submitted to the Swiss Target Prediction webserver (http://www.swisstargetprediction.ch/; accessed on 23 January 2022) [78] and molinspiration webserver (https://www.molinspiration.com/cgi-bin/properties; accessed on 23 January 2022) [79] in order to determine their kinase inhibitory potential and bioactivity scores.

### 4.12. Structure Similarity Comparison

The structure similarity between the six identified hits and co-crystallized ligand structures of DYRK1A and related kinases were studied by ChemMine Web Tools (https://chemminetools.ucr.edu/; accessed on 23 January 2022) [80]. The structures were clustered by hierarchical clustering. The structure similarity of six identified hits were compared with the co-crystallized ligand structures, and distance matrix (Z-scores) were computed. The heatmap visualization of distance matrix was prepared by using GraphPad Prism (version 9.1.0).

### 4.13. Synthetic Accessibility (SA) Prediction

The synthetic accessibility (SA) prediction of the identified molecules was performed by SyntheticAccessibiliyCli (Ambit-SA) software tool (http://ambit.sourceforge.net/reactor.html; accessed on 5 February 2022) [81].

## 5. Conclusions

The discovery of selective and potent inhibitors of DYRK1A is an emerging field of investigation, as decades of research on neurodegenerative diseases, especially Alzheimer’s Disease, have recognized DYRK1A as a novel drug target for therapeutic intervention. In the present study, the mol*Mall* database, consisting of 15,381 rare, diverse and unique compounds, was screened by docking-based virtual screening against a set of five DYR1A protein kinases. Hydrogen bond interaction with Leu241 (gk+3) were considered key determinants for the selection. Binding free energies (ΔG_bind_) of selected ligand–protein complexes were also calculated in order to remove false positive hits. The high scoring selected ligands were further screened for their binding affinity with the other twelve DYRK1A-related protein kinases in order to find selective inhibitors. Among the identified ligands, a bis-indole indigoid scaffold (mol*Mall* ID: 19394) was already reported previously as a DYRK1A inhibitor. Various physicochemical and pharmacokinetic attributions, as well as the toxicity profile of the nominated six hit ligands, were also computed and were found to be satisfactory for orally active CNS drugs. The selected docked ligand–protein complexes were exposed to MD simulations to validate their mechanism of interactions and stability in the ATP pocket of human DYRK1A kinase. The 100 ns simulation time appeared to be suitable, and it is concluded that all the selected ligand structures having *mol*Mall IDs: 9539, 11352, 15938, 19037, 21830 and 21878 are potential inhibitors of the DYRK1A protein.

## Figures and Tables

**Figure 1 molecules-27-01159-f001:**
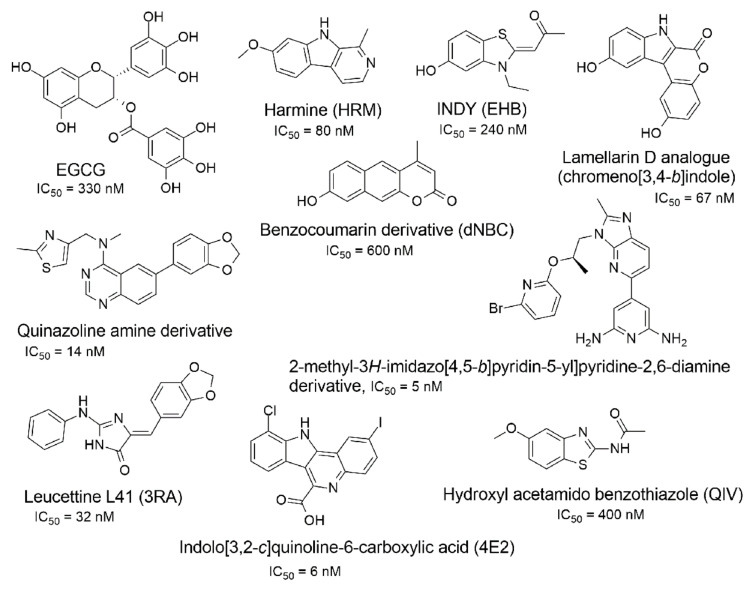
Reported small molecule DYRK1A kinase inhibitors.

**Figure 2 molecules-27-01159-f002:**
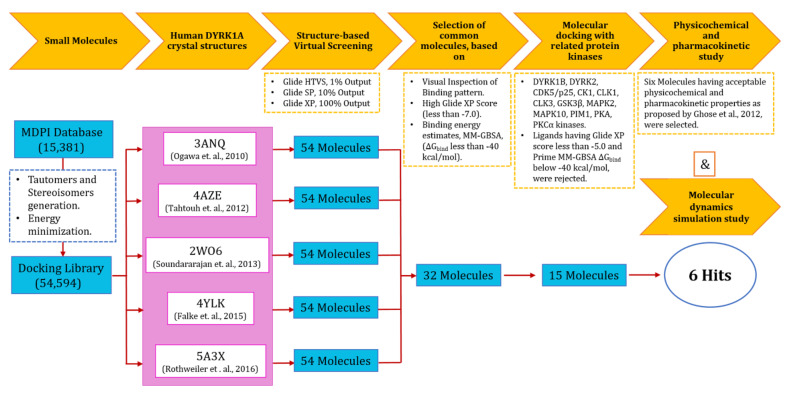
Strategy adopted to identify potential DYRK1A kinase inhibitors.

**Figure 3 molecules-27-01159-f003:**
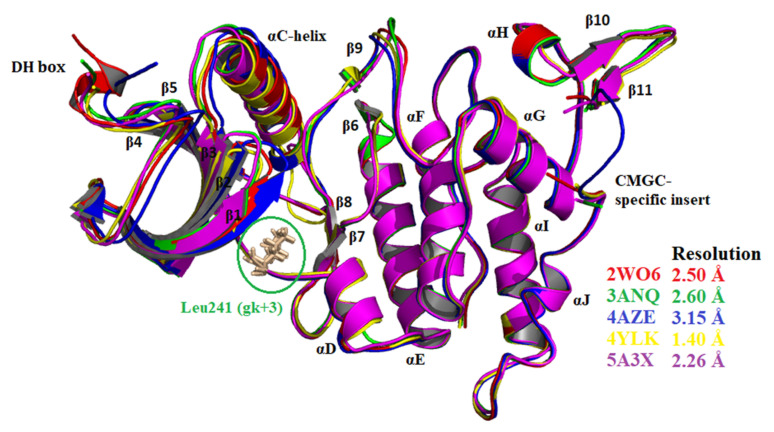
Superposition of five DYRK1A crystal structures, depicting structural variations.

**Figure 4 molecules-27-01159-f004:**
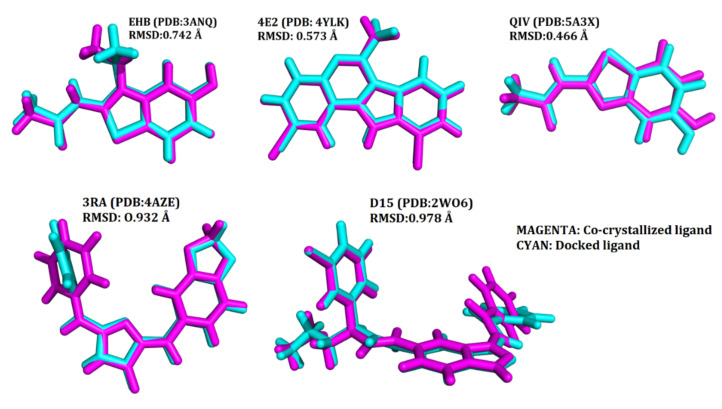
RMSD and poses validating the docking protocol.

**Figure 5 molecules-27-01159-f005:**
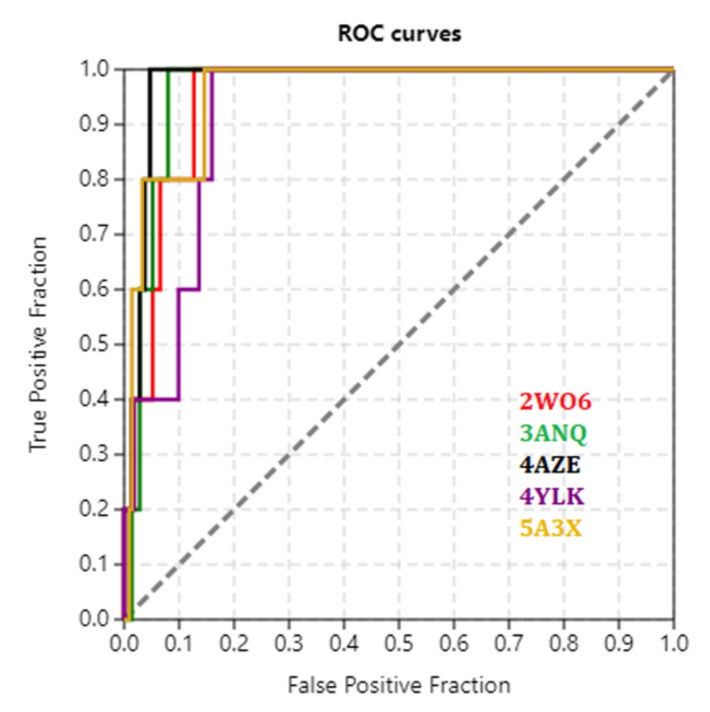
ROC curves depicting docking validation by active decoy-based screening.

**Figure 6 molecules-27-01159-f006:**
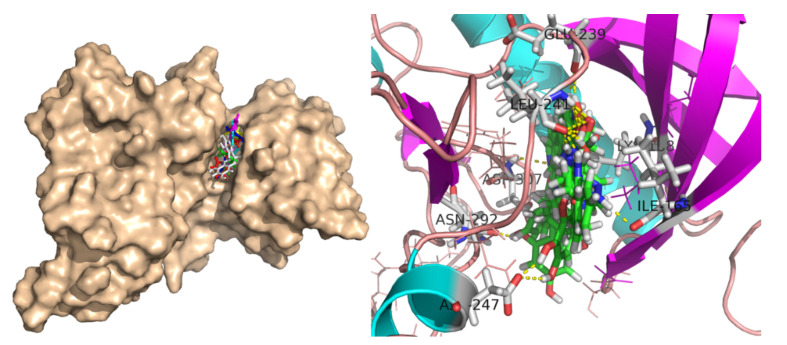
3D binding interactions (surface and cartoon view) of six final selected structures along with reported known inhibitors (EHB, Harmine and EGCG).

**Figure 7 molecules-27-01159-f007:**
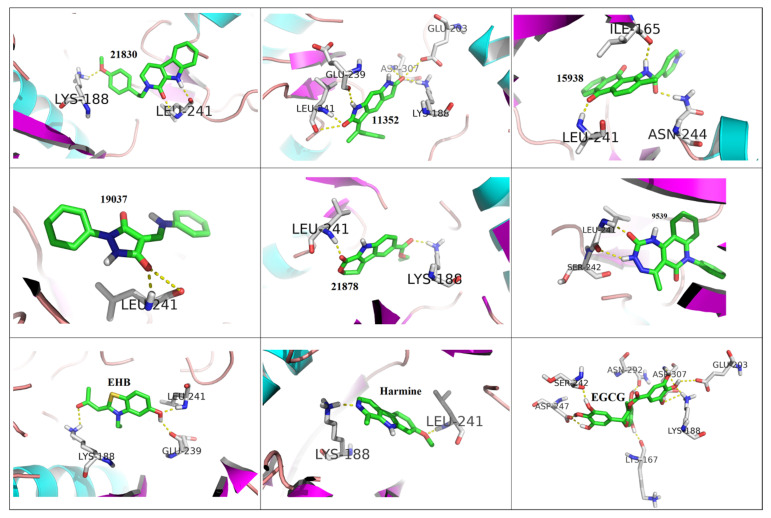
3D binding interactions (cartoon view) of six final selected structures along with reported known inhibitors (EHB, Harmine and EGCG).

**Figure 8 molecules-27-01159-f008:**
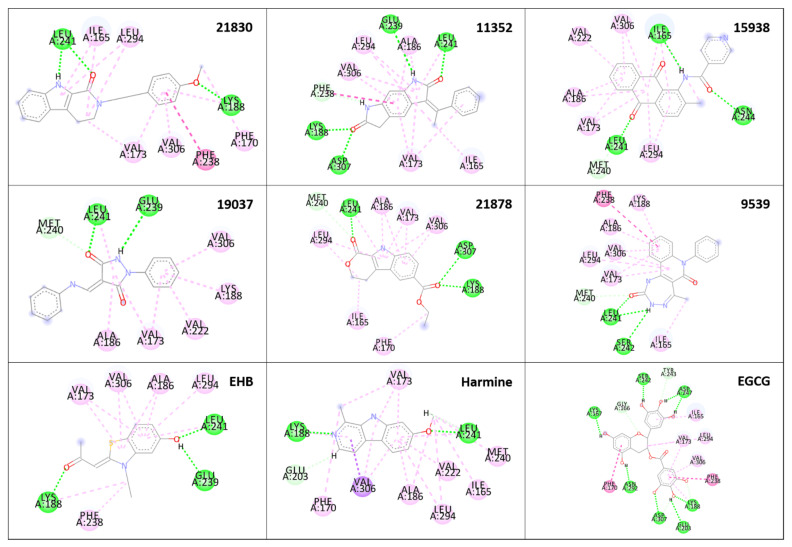
2D binding interactions of six final selected structures along with reported known inhibitors (EHB, Harmine and EGCG).

**Figure 9 molecules-27-01159-f009:**
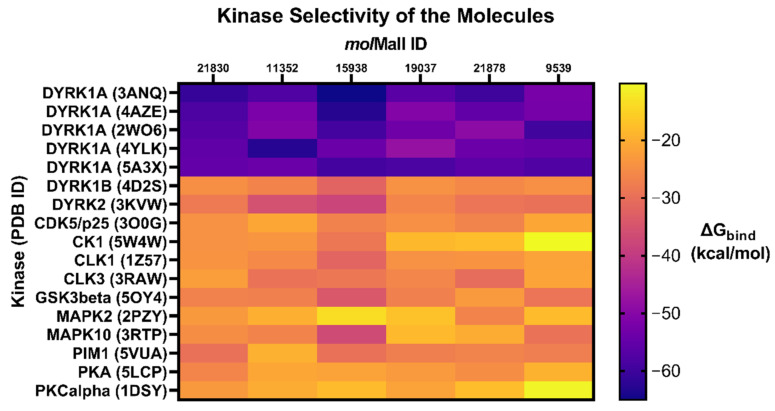
Heatmap analysis of binding energy of six identified molecules with DYRK1A and related kinases.

**Figure 10 molecules-27-01159-f010:**
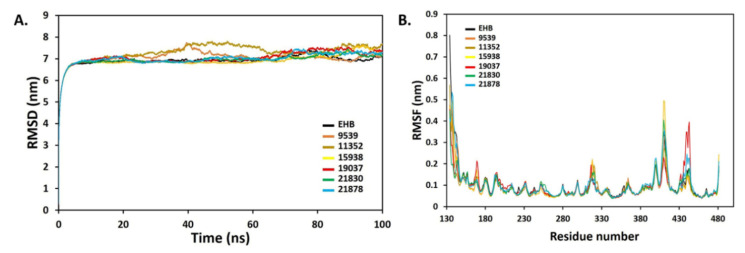
RMSD (**A**) and RMSF (**B**) profile of the protein–ligand complexes of six identified hit ligands (21830, 11352, 15938, 19037, 21878, and 9539), and standard EHB.

**Figure 11 molecules-27-01159-f011:**
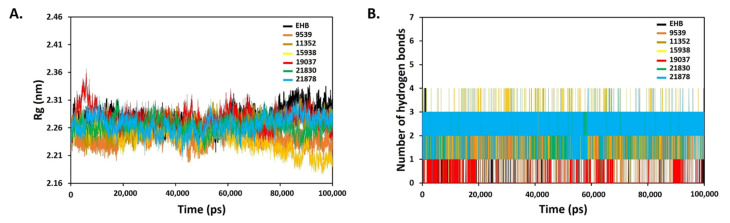
Rg (**A**) and H-bond interactions (**B**) analysis of the protein–ligand complexes of six identified hit ligands (21830, 11352, 15938, 19037, 21878, and 9539), and standard EHB.

**Figure 12 molecules-27-01159-f012:**
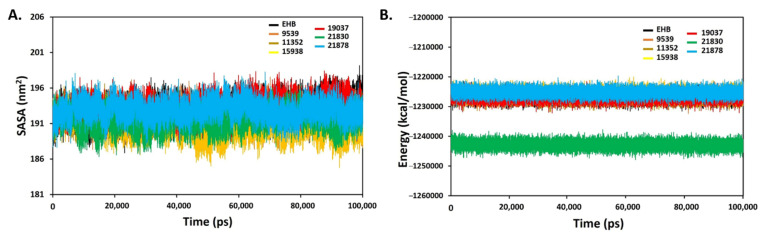
SASA (**A**) and binding energy (**B**) of the protein–ligand complexes of six identified hit ligands (21830, 11352, 15938, 19037, 21878, and 9539), and standard EHB.

**Figure 13 molecules-27-01159-f013:**
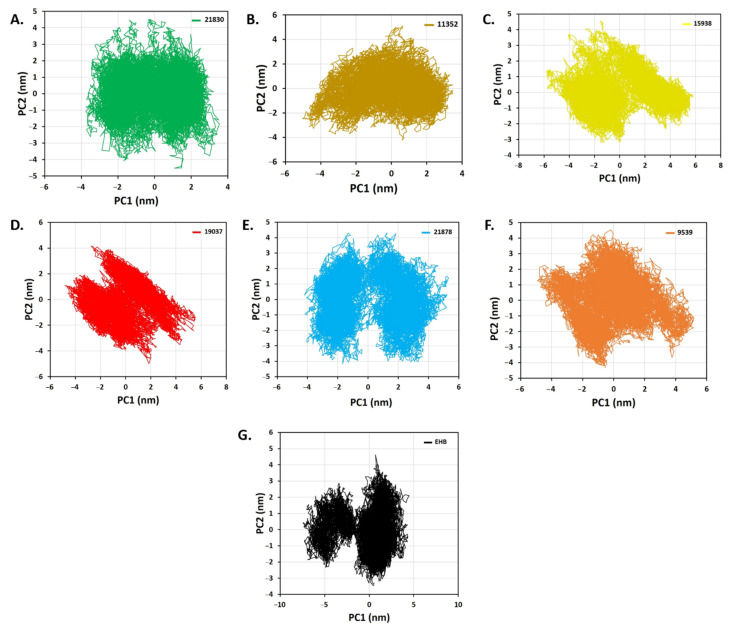
The PCA score plot of the studied protein–ligand complexes of six identified hit ligands (**A**) 21830, (**B**) 11352, (**C**) 15938, (**D**) 19037, (**E**) 21878, (**F**) 9539, and (**G**) EHB (standard).

**Figure 14 molecules-27-01159-f014:**
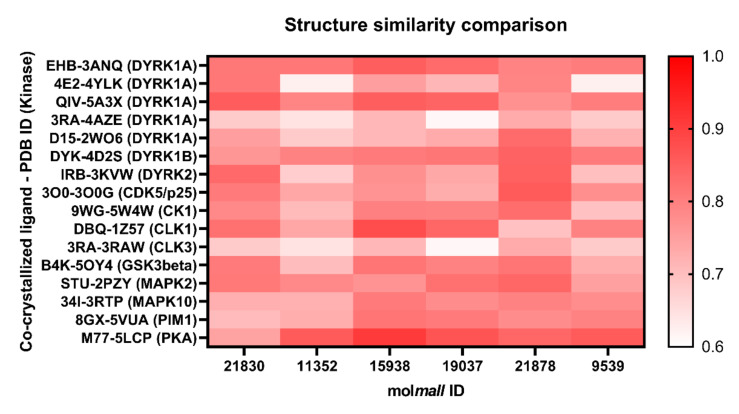
Heatmap visualization of structure similarity comparison between the identified hits and co-crystallized ligands of DYRK1A and related kinases. Lower score indicates structural similarity between the molecules.

**Table 1 molecules-27-01159-t001:** Glide XP docking scores and Prime MM/GBSA binding free energy (kcal/mol) of six identified hit molecules and three known DYRK1A inhibitors.

MDPI (mol*Mall*) ID/Known Inhibitors	Chemical Structures	DYRK1A Crystal Structures (PDB IDs)									
		3ANQ		4AZE		2WO6		4YLK		5A3X	
		Score	ΔG_bind_	Score	ΔG_bind_	Score	ΔG_bind_	Score	ΔG_bind_	Score	ΔG_bind_
21830	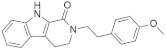	−11.64	−61.15	−10.07	−58.04	−11.23	−56.94	−11.62	−55.68	−11.88	−55.17
11352	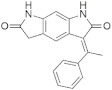	−10.96	−57.79	−11.24	−51.85	−10.80	−50.94	−10.74	−63.00	−10.45	−54.08
15938	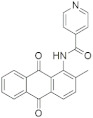	−10.41	−65.02	−10.31	−62.92	−8.47	−59.30	−8.13	−54.20	−8.41	−59.42
19037	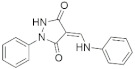	−9.97	−56.42	−8.57	−50.66	−8.51	−53.44	−9.09	−47.94	−10.53	−58.43
21878	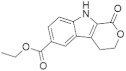	−9.42	−59.96	−10.34	−55.25	−8.34	−49.26	−9.91	−53.93	−8.80	−56.15
9539	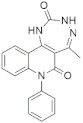	−8.00	−52.33	−10.79	−52.45	−9.08	−59.66	−10.19	−54.87	−10.35	−57.73
EHB	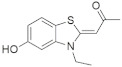	−7.72	−56.42	−7.33	−51.46	−7.49	−54.30	−6.45	−51.68	−7.83	−55.40
Harmine	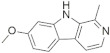	−7.22	−49.24	−6.54	−54.59	−7.42	−46.55	−7.29	−49.11	−8.15	−48.79
EGCG	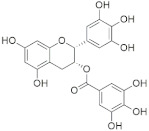	−10.76	−64.44	−12.21	−70.48	−13.17	−63.32	−11.99	−65.67	−11.73	−65.01

Score: Glide XP docking score; ΔGbind: Prime MM/GBSA binding free energy (kcal/mol).

**Table 2 molecules-27-01159-t002:** Molecular contacts profiling for the identified six hit molecules and three known inhibitors in the selective ATP binding pocket of human DYRK1A (PDB ID: 3ANQ).

MDPI (mol*Mall*) ID/Known Inhibitors	Chemical Structures	DYRK1A (PDB ID: 3ANQ): Polar and Non-Polar Interactions		
		H-Bond	Pi-Pi	Alkyl and Pi-Alkyl
21830	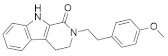	Lys188—OCH_3_ (2.28 Å), Leu241—O=C (2.08 Å), Leu241—NH (2.09)	Phe238	Ile165, Phe170, Val173, Lys188, Leu294, Val306
11352	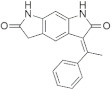	Lys188—O=C (2.14 Å), Glu239—HN (2.52 Å), Leu241—O=C (2.03 Å), Asp307—O=C (2.61 Å)	Phe238	Ile165, Val173, Ala186, Leu294, Val306
15938	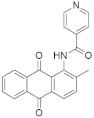	Ile165—HN (2.26 Å), Leu241—O=C (1.99 Å), Asn244—O=C (2.05 Å)	---	Ile165, Val173, Ala186, Val222, Leu294, Val306
19037	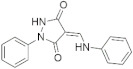	Glu239—HN (2.84 Å), Leu241—O=C (1.91 Å)	---	Val173, Ala186, Lys188, Val222, Leu241, Val306
21878	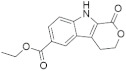	Lys188—O=C (2.13 Å), Leu241—O=C (1.87 Å), Asp307—O=C (2.89 Å)	---	Ile165, Phe170, Val173, Ala186, Leu241, Leu294, Val306
9539	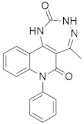	Leu241—O=C (1.86 Å), Leu241—HN (2.44), Ser242—HN (2.98)	Phe238	Ile165, Val173, Ala186, Lys188, Leu294, Val306
EHB	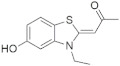	Lys188—O=C (2.72 Å), Glu239—HO (2.15 Å), Leu241—OH (2.00 Å)	---	Val173, Ala186, Lys188, Phe238, Leu241, Leu294, Val306
Harmine	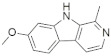	Lys188—N (1.92 Å), Leu241—OCH_3_ (2.13 Å)	---	Ile165, Phe170, Val173, Ala186, Lys188, Val222, Met240, Leu241, Leu294, Val306
EGCG	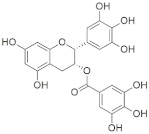	Lys167—HO (1.85 Å), Lys188—OH (1.69 Å), Glu203—HO (2.66 Å), Ser242—HO (1.67 Å), Asp247—HO (1.75 Å), Asp247—HO (1.46 Å), Asn292—HO (1.78 Å), Asp307—OH (2.56 Å)	Phe170, Phe238	Ile165, Val173, Lys188, Leu294, Val306

**Table 3 molecules-27-01159-t003:** Glide XP docking scores and Prime MM/GBSA binding free energy (kcal/mol) of six identified hit molecules after docking with other related protein kinases.

S. No.	Other Protein Kinases (PDB ID)	MDPI (mol*Mall*) ID											
		21830		11352		15938		19037		21878		9539	
		Score	ΔG_bind_	Score	ΔG_bind_	Score	ΔG_bind_	Score	ΔG_bind_	Score	ΔG_bind_	Score	ΔG_bind_
1.	DYRK1B (4D2S)	−3.06	−24.93	−4.41	−26.68	−4.92	−32.29	−1.92	−24.55	−3.67	−26.07	−2.38	−24.76
2.	DYRK2 (3KVW)	−4.41	−28.27	−4.82	−35.88	−4.48	−38.08	−3.33	−26.51	−4.78	−29.17	−1.91	−29.86
3.	CDK5/p25 (3O0G)	−4.66	−24.58	−4.13	−21.07	−4.19	−27.05	−3.56	−24.69	−2.91	−26.82	−2.20	−21.21
4.	CK1 (5W4W)	−4.32	−24.40	−3.56	−23.85	−1.61	−28.72	−3.31	−18.65	−1.91	−17.94	−3.29	−10.20
5.	CLK1 (1Z57)	−3.21	−24.05	−3.25	−25.85	−2.04	−31.94	−3.32	−24.49	−4.52	−24.03	−1.75	−21.87
6.	CLK3 (3RAW)	−3.96	−22.60	−4.72	−29.45	−3.60	−28.74	−3.34	−26.27	−2.75	−30.81	−4.04	−21.52
7.	GSK3β (5OY4)	−3.54	−27.03	−4.47	−27.37	−1.90	−34.49	−3.72	−27.34	−2.36	−23.14	−4.03	−29.32
8.	MAPK2 (2PZY)	−2.68	−23.23	−4.26	−19.91	−2.90	−13.67	−2.01	−17.35	−4.20	−26.60	−1.86	−18.25
9.	MAPK10 (3RTP)	−4.17	−25.13	−4.06	−26.56	−2.97	−37.01	−2.92	−18.56	−3.64	−20.31	−2.69	−29.62
10.	PIM1 (5VUA)	−2.47	−29.84	−3.07	−19.72	−2.46	−29.90	−2.38	−27.41	−2.65	−26.85	−2.86	−27.67
11.	PKA (5LCP)	−2.38	−26.41	−4.46	−21.17	−3.84	−21.99	−1.79	−23.15	−4.50	−25.08	−0.70	−19.48
12.	PKCα (1DSY)	−0.34	−23.23	−3.61	−20.32	−0.37	−18.27	−2.52	−21.98	−2.22	−17.99	0.43	−10.94

Score: Glide XP docking score; ΔG_bind_: Prime MM/GBSA binding free energy (kcal/mol).

**Table 4 molecules-27-01159-t004:** Synthetic accessibility (SA) score of the identified six hits, predicted by Ambit-SA software tool.

mol*Mall* ID	Molecule SMILES	SA Score
21830	c1c2c(ccc1)c1c([nH]2)C(=O)N(CC1)CCc1ccc(cc1)OC	70.672
11352	c1c2c(cc3c1NC(=O)C3)/C(=C(\c1ccccc1)/C)/C(=O)N2	69.201
15938	C1(=O)[C@@H]2[C@H](C(=O)[C@@H]3[C@H]1C=CC=C3)C=CC(=C2NC(=O)c1ccncc1)C	44.679
19037	C1(=O)/C(=C/Nc2ccccc2)/C(=O)N(N1)c1ccccc1	79.648
21878	c1c2c(cc(c1)C(=O)OCC)c1c([nH]2)C(=O)OCC1	72.346
9539	c1c2c(ccc1)n(c(=O)c1c2[nH]c(=O)[nH]nc1C)c1ccccc1	71.309

## Data Availability

The data presented in this study are available on request from the corresponding authors.

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
