# Peer review of "Novel and Potential Small Molecule Scaffolds as DYRK1A Inhibitors by Integrated Molecular Docking-Based Virtual Screening and Dynamics Simulation Study"

_molecules, 2022, doi:10.3390/molecules27041159_

Round 1

Reviewer 1 Report

Dear,

The authors have significantly improved the manuscript in this new version. However, I only ask for a small correction in relation to the formatting of the Tables that were divided, making it difficult to read the data.

In the abstract "The computational toxicity webserver ProTox-II was used to predict the toxicity profile of selected 6 hits" - Cite all six in parentheses.

Make it clear that only the six hits were subjected to the molecular dynamics study and mention which was the best compound or compounds in that sentence "The selected docked ligand-protein systems were exposed to 100 ns molecular dynamics (MD) simulations to validate their mechanism of interactions and stability in the ATP pocket of human DYRK1A kinase".

I firmly believe that the findings reported here will have a major impact on the scientific field, in my opinion I accept for publication the manuscript.

Author Response

Comment-1

The authors have significantly improved the manuscript in this new version. However, I only ask for a small correction in relation to the formatting of the Tables that were divided, making it difficult to read the data.

Response-1

We acknowledge reviewer’s comments and thankful for their valuable suggestions. As per the suggestion, Table 1 & 3 in the manuscript, are now simplified for better understanding and highlighted with red and yellow color.

Comment-2

In the abstract "The computational toxicity webserver ProTox-II was used to predict the toxicity profile of selected 6 hits" - Cite all six in parentheses.

Response-2

In the statement (abstract), all the selected six hit molmall IDs are now included in parentheses and highlighted.

Comment-3

Make it clear that only the six hits were subjected to the molecular dynamics study and mention which was the best compound or compounds in that sentence "The selected docked ligand-protein systems were exposed to 100 ns molecular dynamics (MD) simulations to validate their mechanism of interactions and stability in the ATP pocket of human DYRK1A kinase".

Response-3

As suggested, the information is now included in the abstract and highlighted.

Reviewer 2 Report

Dear authors, based on my previous assessment the concerns were addresses in this resubmitted version. Differences between structures used, docking threshold, improvement of the description of MD parameters have been addressed. The manuscript was improved and, in my opinion, can be accepted for publication.

Author Response

Comment-1

Dear authors, based on my previous assessment the concerns were addresses in this resubmitted version. Differences between structures used, docking threshold, improvement of the description of MD parameters have been addressed. The manuscript was improved and, in my opinion, can be accepted for publication.

Response-1

We are thankful to the reviewers for their valuable suggestions to improve the manuscript.

Reviewer 3 Report

the authors have addressed all my comments

Author Response

Comment-1

The authors have addressed all my comments.

Response-1

We are thankful and indebted to the reviewer for their evaluations and suggestions.

Reviewer 4 Report

This manuscript by Shahroz, M. M. et.al. “Novel and potential small molecule scaffolds as DYRK1A inhibitors by integrated molecular docking-based virtual screening and dynamics simulation study” showed preliminary results of molecular docking virtual screening and simulation study of DYRK1A known inhibitors form the database molecules, it will be useful study for further investigation.

Author Response

Comment-1

This manuscript by Shahroz, M. M. et.al. “Novel and potential small molecule scaffolds as DYRK1A inhibitors by integrated molecular docking-based virtual screening and dynamics simulation study” showed preliminary results of molecular docking virtual screening and simulation study of DYRK1A known inhibitors form the database molecules, it will be useful study for further investigation.

Response-1

We are very much thankful to the reviewers for their effort to improve this manuscript. All the identified six hits will be synthesized, characterized and will be investigated in vitro to confirm their selectivity towards DYRK1A kinase. The planned study/manuscript will be communicated in near future.